# Myelin-Weighted Imaging Presents Reduced Apparent Myelin Water in Patients with Alzheimer’s Disease

**DOI:** 10.3390/diagnostics12020446

**Published:** 2022-02-09

**Authors:** Seung-Hyun Lim, Jiyoon Lee, Sumin Jung, Bokyung Kim, Hak Young Rhee, Se-Hong Oh, Soonchan Park, Ah Rang Cho, Chang-Woo Ryu, Geon-Ho Jahng

**Affiliations:** 1Department of Radiology, Kyung Hee University Hospital at Gangdong, 892 Dongnam-ro, Gangdong-gu, Seoul 05278, Korea; limsh0524@gmail.com (S.-H.L.); bkim0518@naver.com (B.K.); md.cwryu@gmail.com (C.-W.R.); 2Department of Biomedical Engineering, Undergraduate School, College of Electronics and Information, Kyung Hee University, 1732 Deogyeong-daero, Giheung-gu, Yongin-si 17104, Korea; little8867@naver.com (J.L.); jungsm0411@khu.ac.kr (S.J.); 3Department of Neurology, Kyung Hee University Hospital at Gangdong, 892 Dongnam-ro, Gangdong-gu, Seoul 05278, Korea; azzo73@gmail.com; 4Department of Medicine, College of Medicine, Kyung Hee University, 26 Kyung Hee Dae-ro, Dongdaemun-gu, Seoul 02447, Korea; mdpark96@gmail.com (S.P.); ppirru@naver.com (A.R.C.); 5Division of Biomedical Engineering, Hankuk University of Foreign Studies, Yongin 17035, Korea; jakeoh79@gmail.com; 6Department of Psychiatry, Kyung Hee University Hospital at Gangdong, 892 Dongnam-ro, Gangdong-gu, Seoul 05278, Korea

**Keywords:** Alzheimer’s disease, myelin water imaging, myelin loss, clinical protocol

## Abstract

The purpose of this study was to investigate myelin loss in both AD and mild cognitive impairment (MCI) patients with a new myelin water mapping technique within reasonable scan time and evaluate the clinical relevance of the apparent myelin water fraction (MWF) values by assessing the relationship between decreases in myelin water and the degree of memory decline or aging. Twenty-nine individuals were assigned to the cognitively normal (CN) elderly group, 32 participants were assigned to the MCI group, and 31 patients were assigned to the AD group. A 3D visualization of the short transverse relaxation time component (ViSTa)-gradient and spin-echo (GraSE) sequence was developed to map apparent MWF. Then, the MWF values were compared between the three participant groups and was evaluated the relationship with the degree of memory loss. The AD group showed a reduced apparent MWF compared to the CN and MCI groups. The largest AUC (area under the curve) value was in the corpus callosum and used to classify the CN and AD groups using the apparent MWF. The ViSTa-GraSE sequence can be a useful tool to map the MWF in a reasonable scan time. Combining the MWF in the corpus callosum with the detection of atrophy in the hippocampus can be valuable for group classification.

## 1. Introduction

In a nervous system, a neuron is a cell that sends signals to other cells through fragile and thin axons while a glial cell forms a membranous sheath called myelin, which surrounds and eventually insulates, axons [1]. The oligodendrocytes and schwann cells are specialized glial cells that produce the myelin sheaths of the central nervous system and peripheral nervous system, respectively. The Oligodendrocytes myelinate multiple axons in the central nervous system, while Schwann cells myelinate a single axon in the peripheral nervous system. Myelinated axons appear as spiral wrappings of the myelin sheath. Demyelination is the condition in which preexisting myelin sheaths are damaged without axonal damage. It can result from various medical conditions such as viral infection, inflammatory process, metabolic dysfunctions. [2,3]. Myelin loss can cause nerve dysfunction by slowing or stopping nerve conduction through the axons. Myelin proteins are reduced with age [4] and further reduced in mild cognitive impairment (MCI) and Alzheimer’s disease (AD) [5,6,7]. In addition, myelin pathology contributes to the decline of cognitive characteristics in patients with AD [8]. Previous biological studies in AD brains reported that decreased myelin basic protein, myelin proteolipid protein, and cholesterol levels in white matter [7,9] and the focal loss of myelin were associated with amyloid-beta (Aβ) plaques in the gray matter [10] and the white matter [11] of AD patients. Therefore, the investigation of demyelination in the AD brain could be helpful in understanding AD. Developing an imaging biomarker, which could indicate myelin loss, would have important clinical implications for the diagnosis and prognosis of diseases.

Several methods have been introduced to map the myelin water fraction (MWF) using MRI [12]. Most methods have been based on the measurement of T2 or T2* relaxation using a multi-echo sequence to fit multi-exponential relaxation times [13,14,15,16,17]. There are drawbacks to the multi-components fitting methods, including a long scan time (10~30 min) to obtain multi-echo images and unreliability to fit a short component due to noise. Therefore, those fitting methods may be inapplicable to clinical routine studies, especially in AD patients. To improve the signal-to-noise ratio, reduce the scan time, and ignore fitting errors, a T1-based method called the visualization of short transverse relaxation time component (ViSTa) was introduced [18,19,20]. ViSTa suppresses long T1 signals from cerebrospinal fluid and white matter and gray matter using optimized double-inversion times. Therefore, the signal from a short T1 component in the white matter can be detected using a ViSTa preparation period. The previous multi-component fitting methods can be used to map MWF quantitatively, but the ViSTa-based method can map the apparent MWF. Therefore, in this paper, the term apparent MWF was used.

Since some studies reported myelin changes in AD patients, we thought that it is important to develop an imaging technique to map myelin changes in the brain within a reasonable imaging time in clinics, especially in AD patients who experience continuous subtle movements due to the inability to follow orders. Therefore, the objectives of this study were: (1) to investigate myelin loss in participants with AD and amnestic MCI compared to cognitively normal (CN) elderly individuals using a whole-brain MWF map and (2) to investigate the clinical relevance of the apparent MWF by assessing the relationship between reductions in myelin water and the degree of memory loss.

## 2. Materials and Methods

### 2.1. Participants

Our Institutional Review Board (IRB) approved this cross-sectional prospective study. Informed consent was obtained from the participants. The participants provided a detailed medical history and underwent a neurologic examination, standard neuropsychological testing, and MRI scanning. A standard neuropsychological test, the Seoul Neuropsychological Screening Battery [21] from the Korean version of the Mini-Mental State Examination (MMSE) for global cognitive ability was used to evaluate cognitive function. Based on the results of the neuropsychological examination and MRI findings, the participants were categorized into three groups. First, according to the criteria of the National Institute of Neurological and Communicative Disorders and Stroke-Alzheimer Disease and Related Disorders Association, patients with Clinical Dementia Rating (CDR) scores of 0.5, 1, or 2 were classified as mild and probable AD. [21]. Secondly, the participants with amnestic MCI were also included according to the Petersen criteria [22]. Finally, elderly CN participants were included from healthy volunteers with no medical history of a neurological disease. This study included a total of 105 participants. Among them, 13 participants were excluded from the subsequent analysis. Therefore, the remaining 92 participants were allocated to the CN elderly (*n* = 29), the amnestic MCI (*n* = 32), and the AD (*n* = 31) groups. Table 1 summarizes the demographic characteristics of the participants.

### 2.2. MRI Acquisition

To calculate the MWF value in each voxel, two whole-brain images were acquired using a 3D gradient and spin-echo (GraSE) sequence [23] with and without the ViSTa preparation [18,19] using a 32-channel sensitivity-encoding (SENSE) array coil. At first, the GraSE sequence was run with the ViSTa preparation part. Two inversion times (TI) of 742 ms and 222 ms were used for suppressing the long T1 components and fat signals. The following imaging parameters were used: repetition time (TR)/echo time (TE) =1160/11 ms, flip angle (FA) = 90°, acquisition matrix = 112 × 89, acquisition voxel size = 2 × 2 × 5 mm, reconstruction matrix size = 1 × 1 × 5 mm, slice oversampling factor = 1.5, SENSE factor = 2 for the phase reduction direction and 1 for the slice reduction direction, turbo spin-echo (TSE) factor = 9 with low-high k-space profile order, echo-planar imaging (EPI) factor = 3, number of slices = 25, number of average = 2, fat suppression = SPIR, and imaging orientation = transverse with a right-left fold-over direction. The scan time was 2 min 35 s. In the ViSTa-GraSE sequence, one-echo images were acquired, which was different from other myelin water imaging methods. Second, the GraSE sequence was run again with the same imaging parameters, but without using the ViSTa preparation part for acquiring a reference image to quantify the apparent MWF in each voxel. The scan time was only 28 s.

The sagittal structural 3D T1-weighted (3D T1W) images were acquired for image registration and brain tissue segmentation. At the same time, the turbo field echo sequence which is similar to the magnetization-prepared rapid acquisition of the gradient echo (MPRAGE) sequence were also acquired. The following parameters were applied: TR = 8.1 ms, TE = 3.7 ms, FA = 8°, field-of-view (FOV) = 236 × 236 mm^2^, and voxel size = 1 × 1 × 1 mm^3^. Finally, T2-weighted turbo-spin-echo and fluid-attenuated inversion recovery (FLAIR) images were acquired to evaluate any brain abnormalities. The imaging parameters of the T2-weighted images were as follows: TR/TE = 3786/80 ms, slice thickness = 5 mm, acquisition matrix size = 460 × 368, and FOV = 179 × 230, obtained in the coronal direction. The imaging parameters of the FLAIR were as follows: TR/TI = 11,000/2800 ms, TE = 125 ms, acquisition voxel size (MPS) = 0.65 × 0.96 × 3.00 mm, slice thickness = 3 mm, slice gap = 0.3 mm, and number of slices = 45, obtained in the transverse direction. MRI was performed using a 3.0 Tesla MRI system (Ingenia, Philips Medical System, Best, The Netherlands).

### 2.3. Post-Processing for Mapping MWF

We performed the following steps using Statistical Parametric Mapping version 12 (SPM12) software (http://www.fil.ion.ucl.ac.uk/spm/software/spm12/ (accessed on 20 August 2020)). To create the apparent MWF map, the ViSTa-GraSE image and the corresponding reference image were co-registered to minimize the motion between the scans. The apparent MWF map for each voxel was created from the ViSTa-GraSE image divided by the reference image and multiplied by 100% and a scaling factor. The scaling factor was 0.763561 which was considered to be the difference of TR and TE for the ViSTa-GraSE image and the corresponding reference image and the exponential signal decay effects with T1 and T2* relaxation times. [19]. Then, to evaluate the difference in the apparent MWF map within the three groups, the 3D T1W image and the reference image for each subject were co-registered. As a result, the MWF map was also co-registered. Then, the 3D T1W image was segmented into gray matter and white matter using the CAT12 tool (http://www.neuro.uni-jena.de/cat/ (accessed on 24 September 2020)) to obtain brain tissue information and was spatially normalized to the standard dementia brain template generated by our lab [24]. The apparent MWF map was normalized into the dementia standard template using the deformation field information of the 3D T1W. Finally, Gaussian smoothing using a full-width at half maximum (FWHM) of 10 × 10 ×10 mm^3^ was performed for the voxel-based statistical analysis of all maps, including the gray matter and white matter volumes.

### 2.4. Statistical Analysis

#### 2.4.1. Demographic Data and Clinical Outcome Scores

The demographic data and clinical outcome scores were compared between the three participant groups. One-way analysis of variance (ANOVA) was used to compare age and MMSE scores between the participant groups. Sex was compared using the chi-squared test.

#### 2.4.2. Voxel-Based Analysis of Apparent MWF Maps

We performed two voxel-based analyses. First, the voxel-based full factorial ANCOVA test was used for group comparisons of the apparent MWF maps with participants’ age as a covariate. In addition, the same analyses were performed for both gray matter volume (GMV) and white matter volume (WMV) with participants’ total intracranial volume (TIV), age, and sex as covariates. Second, voxel-based multiple regression analysis was performed to evaluate the relationship between the apparent MWF maps and age without using any covariate, and between the MWF maps and the MMSE scores with participant’s age as a covariate. In addition, the same analyses were performed to evaluate the relationship between the apparent GMV and WMV maps and age with participant’s TIV and sex as covariates, and between the GMV and WMV maps and the MMSE scores with participants’ TIV, sex, and age as covariates. A significance level of α = 0.001 was applied with correction for multiple comparisons using the false discovery rate (FDR) method and clusters with at least 100 contiguous voxels.

#### 2.4.3. Region-of-Interest (ROI)-Based Analysis of Apparent MWF Values

The ROIs were defined in the areas of the corpus callosum, cingulate gyrus, cingulum, hippocampus, middle temporal gyrus, parahippocampal gyrus, pons, precuneus, and thalamus based on the results of the voxel-based analysis and based on the knowledge of the affected locations in AD patients. The atlas-based ROIs were defined using wfu_pickatlas software (http://fmri.wfubmc.edu/software/PickAtlas (accessed on 7 September 2021)). The apparent MWF, GMV, and WMV values were extracted from the selected ROIs.

We performed the following statistical analyses using the ROI data. First, an ANOVA test was performed to evaluate the differences in ROI values between the three participant groups. If there was any significant difference between the groups, then the Scheffe multiple comparison test was used as the post hoc test. Second, Pearson’s correlation coefficient test was performed to analyze the degree of association between the ROI values and participants’ age. Furthermore, we performed a partial correlation analysis between the ROI values and MMSE scores with adjustment for participants’ age. Because of the negative correlation between MWF and age, we analyzed the partial correlation with an adjusted age. Finally, a receiver operating characteristic (ROC) curve analysis was used to evaluate the differences between the two groups for each ROI value, including the apparent MWF, GMV, and WMV. The additional ROC curve analysis was also performed to evaluate the improvement in the group classification by adding brain tissue volumes to the apparent MWF value. For the ROI analysis, α < 0.05 was used to determine the significance level. The statistical analysis was performed using the Medcalc (MedCalc Software, Acacialaan, Ostend, Belgium) statistical program.

## 3. Results

### 3.1. Participant Characteristics

Age (F = 1.586, *p* = 0.210) was not significantly different between the participant groups, but the MMSE score (F = 63.375, *p* < 0.001) was significantly different between AD and other groups, as expected. Sex (χ^2^ < 3.673, *p* > 0.055) did not differ significantly between the three participant groups. The results of the statistical analysis of the participant’s demographic data and the neuropsychologic tests are summarized in Table 1.

### 3.2. Voxel-Based Analysis of Apparent MWF Maps

Figure 1 shows the representative maps of the apparent MWF with and without using color, 3D T1W, GMV, and WMV obtained from one elderly CN participant (72-year-old female), one MCI (72-year-old female) participant, and one AD (73-year-old female) participant. The apparent MWF signal was nicely depicted around the white matter areas. A gross signal drop in the apparent MWF signal was observed in the AD group compared to the CN group in the white matter.

Figure 2 shows the results of the voxel-based ANCOVA analysis of the apparent MWF, GMV, and WMV maps among the three participant groups. The apparent MWF was reduced in the AD group at most of the brain areas included in the cingulate gyrus, the temporal lobe, and the parietal lobe compared to the CN and MCI groups. Both GMV and WMV were also decreased in the AD group. In addition, those values were decreased in the MCI group compared to the CN group. The detailed locations of the significant differences between the participant groups are listed in Appendix A for the apparent MWF, Appendix A for GMV, and Appendix A for WMV.

The red color indicates greater measures in the participant group, indicated in each comparison.

Figure 3 shows the results of voxel-based multiple regression analysis of the apparent MWF, GMV, and WMV maps using the age of all participants. MWF was decreased with increasing age at most of the brain areas. Both GMV and WMV were also decreased with increasing age. However, we did not find any association between MWF, GMV, and WMV and the MMSE scores in the voxel-based analysis using the whole cohort. In addition, we did not find any association between MWF and the MMSE scores in each participant group. The detailed locations of the significantly associated areas are listed in Appendix A for the apparent MWF, Appendix A for GMV, and Appendix A for WMV.

The significant negative associations between the MRI measures and age are shown in blue color.

### 3.3. ROI-Based Analysis of the Apparent MWF Maps

ANOVA tests: Table 2 summarizes the mean values with standard deviations for the apparent MWF, GMV, and WMV for each ROI. The apparent MWF values in all ROIs were significantly different between the three participant groups (F > 4, *p* < 0.009). The MWF values in all ROIs were decreased in the AD group compared to the CN and MCI groups but were not significantly different between the CN and MCI groups, which was the same as the result of the voxel-based comparison. Both GMV and WMV in all ROIs were also significantly decreased in the AD group compared to the CN and MCI groups.

#### 3.3.1. Correlation with Age and MMSE Scores

Table 3 lists the results of the correlation analysis between the MRI measures and age or MMSE scores in all ROIs. The apparent MWF values in all ROIs were significantly negatively correlated with age (Figure 4). Both the GMV and WMV values were also significantly negatively correlated with age. With adjustment for age, the apparent MWF values were significantly positively correlated with the MMSE scores in most ROIs, except for the cingulum and middle temporal gyrus (Table 3 and Figure 5). In addition, both the GMV and WMV values were also significantly positively correlated with the MMSE scores.

#### 3.3.2. ROC Curve Analyses

The ROC curve analysis results for each MRI measure in each ROI are summarized in Appendix A. The apparent MWF values in all ROIs significantly differed between the CN and AD groups and between the MCI and AD groups. For classification between the CN and AD groups, the largest AUC for each MRI measure was 0.799 in carpus callosum for MWF, 0.883 in the hippocampus for GMV, and 0.860 in the hippocampus for WMV. For classification between the MCI and AD groups, the largest AUC for each MRI measure was 0.799 in the carpus callosum for MWF, 0.771 in the hippocampus for GMV, and 0.740 in the hippocampus for WMV. The AUC value was lower in differentiating MCI from AD for all three MRI measures compared with that for classification between other groups. We selected MWF in the carpus callosum and GMV and WMV in the hippocampus for evaluating the added value of MWF.

Table 4 lists the results of the ROC curve analysis to evaluate adding MWF to the GMV and/or WMV values for group classification. To differentiate between the AD patients and the CN participants, the largest AUC was seen by adding all three MRI measures (AUC = 0.905), and the second-largest AUC was seen by adding MWF to GMV (AUC = 0.898) or to WMV (AUC = 0.891). To differentiate between the AD patients and the MCI participants, the largest AUC was seen by adding all three MRI measures (AUC = 0.812), and the second-largest AUC was seen by adding MWF to GMV (AUC = 0.803) or to WMV (AUC = 0.794). The addition of the MWF value of the corpus callosum areas and the GMV values of the hippocampus (SE = 86, SP = 80, AUC = 0.883) slightly improved the discrimination between the CN and AD groups (SE = 93, SP = 77, AUC = 0.898). However, there were no significant differences of ROC areas in any parameter combinations.

## 4. Discussion

### 4.1. Apparent MWF Signals in AD

A new 3D ViSTa-GraSE sequence was employed to evaluate the routine clinical application of the apparent MWF in AD patients who usually have difficulty keeping their heads motionless during an MRI scan. The results showed that the MWF was significantly reduced in the AD group compared to the CN and MCI groups and decreased with disease severity (Table 2). The MWF values provided high AUC values in the corpus callosum (SE = 86, SP = 67, AUC = 0.799) to differentiate the AD patients from the CN participants (Appendix A Appendix A). Furthermore, the addition of the MWF value of the corpus callosum areas and the GMV values of the hippocampus (SE = 86, SP = 80, AUC = 0.883) slightly improved the discrimination between the CN and AD groups (SE = 93, SP = 77, AUC = 0.898) (Table 4). Therefore, the MWF of the corpus callosum area along with the detection of atrophy of the hippocampus can be valuable for improving the classification of individuals.

This study showed a significant reduction in the MWF in AD patients compared to the CN participants. Previous imaging studies with a T2-based multi-echo sequence also demonstrated substantially decreased MWF values in AD patients including the medial temporal lobes, parietal lobes, splenium, and genu of the corpus callosum, and the WM surrounding the frontal and posterior aspect of the lateral ventricles [25,26,27,28]. Previous postmortem studies also showed that MCI and AD patients showed more severe myelin damage than healthy elders due to unsuccessful repair processes [6,7,11,29,30,31,32,33]. AD patients showed decreased myelin basic protein and myelin proteolipid protein, as well as a significant reduction in white matter cholesterol levels [7,9]. The results of the current study confirm the findings of older ones, not the other way. Moreover, since a previous study showed that the greatest degree of demyelination occurred at the plaque core in both human AD and mouse model [10], future studies scanning both VisTa-GraSE and amyloid and/or tau PET are recommended to verify the association of MWF reduction and depositions of Aβ and/or phosphorylated tau in AD.

### 4.2. Apparent MWF Signals Correlate with MMSE Scores and/or Age

The apparent MWF value was decreased with increasing age in all ROIs (Table 3). This is consistent with the age-related changes recently observed in healthy subjects using a T2-based multi-echo sequence [25,26,28]. The previous study showed that myelin was increased up to around 40 years and then gradually decreased with increasing age.

Although the voxel-based analysis did not show any association between MWF and MMSE scores, the ROI-based analysis showed that the apparent MWF value in most ROIs was significantly reduced as the MMSE scores decreased, except cingulum and middle temporal gyrus (Table 3). A previous study also showed positive relationships between MWF and MMSE scores [34], indicating that the memory impairment in aging, MCI, and AD patients was related to myelin breakdown [29,35,36]. Furthermore, neuronal AD pathology, which is characterized by neuronal and axonal dysfunction, could alter the amount of myelin produced by oligodendrocytes. Therefore, the positive relationship between MWF and MMSE can be explained by disease progression. The MWF imaging biomarker can be used to track the dynamic changes of the brain myelin as cognition declines.

### 4.3. Limitations

This study had several limitations. First, we did not directly compare the MWF maps with those obtained with the multi-echo T2 relaxation fitting method. Furthermore, we did not compare the results with the autopsy results. In addition, the MWF maps obtained with ViSTa have not yet been histologically validated in AD patients. Therefore, further studies should be performed before being routinely used in clinics. Second, this study was performed as a cross-sectional study. Therefore, a longitudinal study is recommended to evaluate the rate of demyelination in AD patients. Consequently, the number of participants in each group was small and, therefore, future studies are required with larger sample sizes.

## 5. Conclusions

The 3D ViSTa-GRASE sequence provided a good MWF map in a reasonable scan time to evaluate myelin loss in AD patients. MWF was significantly reduced in AD patients compared to the CN participants in most of the white matter areas, especially in the corpus callosum in the brain, indicating the presence of widespread demyelination in AD patients. In addition, MWF was significantly correlated with memory loss. Consequently, for evaluating demyelination and predicting treatment outcomes in AD, MWF could serve as a potential imaging biomarker. The differentiation of AD patients from the CN participants using both GMV and MWF was high, indicating its potential for clinical use. MWF in the corpus callosum areas with atrophy detection in the hippocampus can be valuable for improving the classification of individuals as a promising imaging biomarker.

## Figures and Tables

**Figure 1 diagnostics-12-00446-f001:**
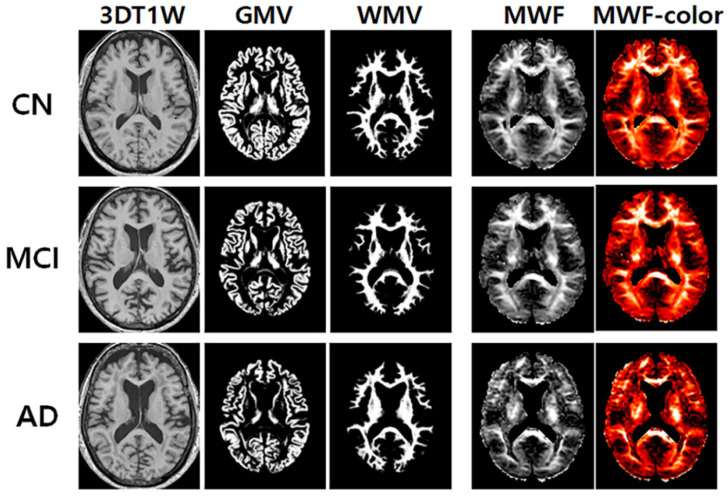
Representative maps of the three-dimensional (3D) T1-weighted (T1W), gray matter volume (GMV), white matter volume (WMV), and apparent myelin water fraction (MWF) with and without color code.

**Figure 2 diagnostics-12-00446-f002:**
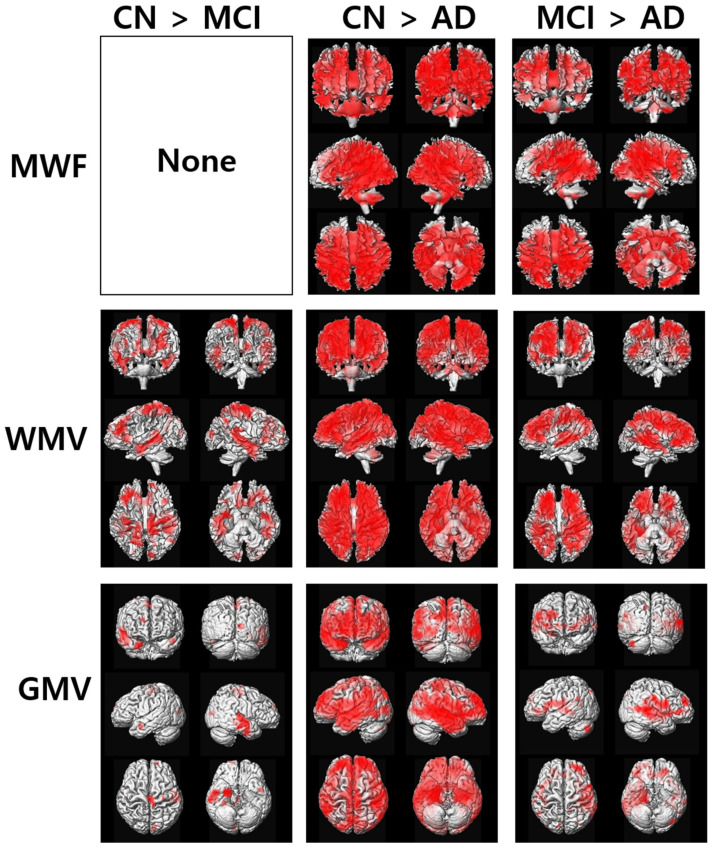
Results of voxel-based ANCOVA analysis of the maps of the apparent myelin water fraction (MWF), gray matter volume (GMV), and white matter volume (WMV) in the three participant groups of cognitively normal (CN) elderly individuals, participants with mild cognitive impairment (MCI), and patients with Alzheimer’s disease (AD).

**Figure 3 diagnostics-12-00446-f003:**
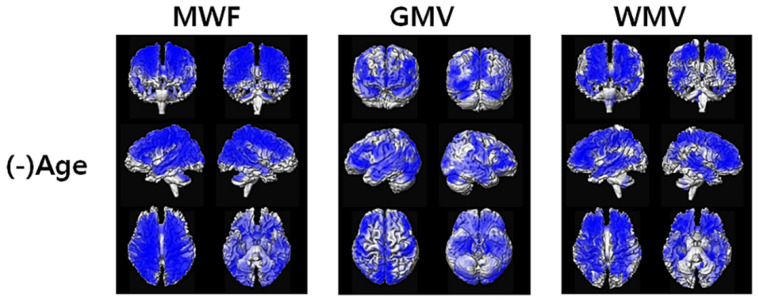
Results of the voxel-based multiple regression analysis between the all-participant maps of the apparent myelin water fraction (MWF), gray matter volume (GMV), and white matter volume (WMV).

**Figure 4 diagnostics-12-00446-f004:**
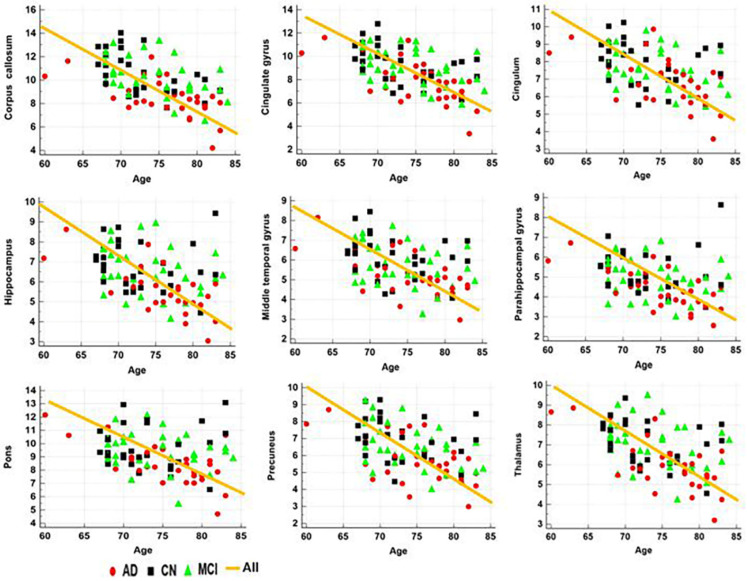
Results of the correlation analysis between the apparent MWF values in the specific brain areas and age.

**Figure 5 diagnostics-12-00446-f005:**
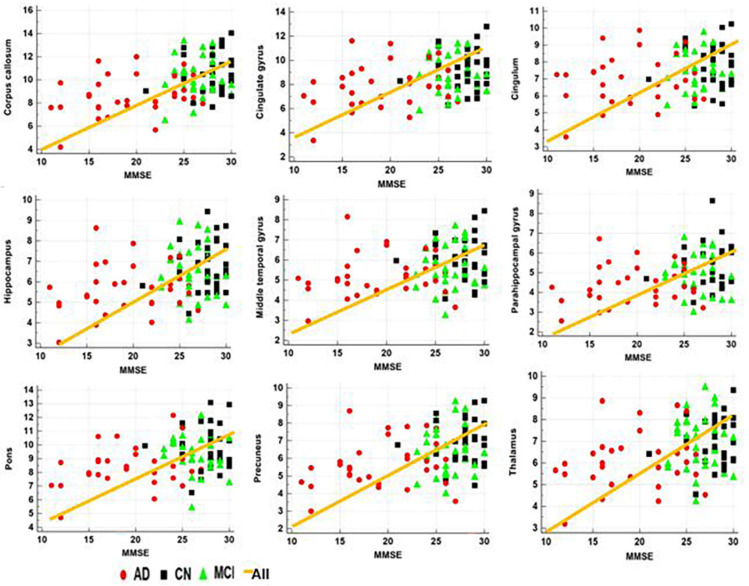
Results of the correlation analysis between the apparent MWF values in the specific brain areas and MMSE scores.

**Table 1 diagnostics-12-00446-t001:** Summary of the statistical results of the demographic data.

Group	CN(1)	MCI(2)	AD(3)	Statistical Tests (Post Hoc)
No. of participants	29	32	31	92 (total)
Demographic data and neuropsychologic test				
Age (year) *	73.00 ± 5.01	74.31 ± 5.02	75.42 ± 5.71	F = 1.586/*p* = 0.210
Sex # (male/female)	13/16 (M:44.8% F:55.2%)	11/21 (M:34.4% F:65.6%)	5/26 (M:16.13% F:83.87%)	χ^2^ < 3.673/*p* > 0.055
MMSE * (/30)	27.93 ± 2.02	26.53 ± 1.90	19.39 ± 4.74	F = 63.375/*p* < 0.001 (1, 3), (2, 3)
CDR(range)	0 (0–0.5)	0.5 (0.5)	1 (0.5–2)	N/A

The data are listed as the mean ± standard deviation for the continuous variables. Clinical Dementia Rating (CDR) scores are presented as the median (range) values. * ANOVA test (F/*p* values*) comparing the continuous variables between the three groups. After the ANOVA test, the Scheffé post hoc test was performed and the results reflect the significant difference between the three subject groups (1, 2, 3), between the CN and AD groups (1, 3), between the CN and MCI groups (1, 2), and between MCI and AD group (2, 3). # *p*-value by the chi-squared test.

**Table 2 diagnostics-12-00446-t002:** Results of comparisons of MRI measures between the three participant groups in specific brain areas.

ROI	MRI Measures	CN(1)	MCI(2)	AD(3)	ANOVA, F/*p*-Value * (Post Hoc)
Corpus callosum	MWF	10.60 ± 7.80	10.31 ± 1.65	8.59 ± 1.70	F = 12.204/*p* < 0.001 (1, 3), (2, 3)
	GMV	0.12 ± 0.013	0.11 ± 0.01	0.11 ± 0.01	F = 2.982/*p* = 0.056
	WMV	0.42 ± 0.06	0.39 ± 0.04	0.36 ± 0.05	F = 9.711/*p* < 0.001 (1, 3), (2, 3)
Cingulate gyrus	MWF	9.34 ± 1.60	9.10 ± 1.55	7.88 ± 1.81	F = 6.718/*p* = 0.002 (1, 3), (2, 3)
	GMV	0.26 ± 0.03	0.24 ± 0.02	0.23 ± 0.03	F = 6.044/*p* = 0.003 (1,3)
	WMV	0.36 ± 0.05	0.33 ± 0.04	0.31 ± 0.04	F = 12.562/*p* < 0.001 (1, 3), (2, 3)
Cingulum	MWF	7.88 ± 1.29	7.66 ± 1.17	6.92 ± 1.42	F = 4.673/*p* = 0.012 (1, 3)
	GMV	0.34 ± 0.04	0.32 ± 0.03	0.30 ± 0.04	F = 9.970/*p* < 0.001 (1, 2), (1, 3)
	WMV	0.26 ± 0.04	0.24 ± 0.03	0.22 ± 0.03	F = 12.661/*p* < 0.001 (1, 2, 3)
Hippocampus	MWF	6.83 ± 1.18	6.56 ± 1.19	5.64 ± 1.19	F = 8.269/p = 0.001 (1, 3), (2, 3)
	GMV	0.37 ± 0.05	0.33 ± 0.05	0.28 ± 0.04	F = 20.880/*p* < 0.001 (1, 2, 3)
	WMV	0.26 ± 0.03	0.25 ± 0.03	0.22 ± 0.03	F = 16.568/*p* < 0.001 (1, 3), (2, 3)
Middle temporal gyrus	MWF	6.18 ± 1.05	5.78 ± 1.09	5.21 ± 1.05	F = 6.250/*p* = 0.003 (1, 3)
	GMV	0.33 ± 0.04	0.32 ± 0.03	0.29 ± 0.04	F = 13.417/*p* < 0.001 (1, 3), (2, 3)
	WMV	0.20 ± 0.02	0.18 ± 0.02	0.16 ± 0.02	F = 13.223/*p* < 0.001 (1, 3),(2, 3)
Parahippocampal gyrus	MWF	5.35 ± 1.08	4.97 ± 0.95	4.34 ± 0.97	F = 7.914/*p* = 0.001 (1, 3), (2, 3)
	GMV	0.41 ± 0.05	0.38 ± 0.05	0.34 ± 0.05	F = 18.772/*p* < 0.001 (1, 2, 3)
	WMV	0.21 ± 0.02	0.19 ± 0.02	0.17 ± 0.03	F = 16.048/*p* < 0.001 (1, 3), (2, 3)
Pons	MWF	9.74 ± 1.60	9.46 ± 1.36	8.47 ± 0.48	F = 6.143/*p* = 0.003 (1, 3), (2, 3)
	GMV	0.04 ± 0.004	0.03 ± 0.004	0.03 ± 0.004	F = 6.778/*p* = 0.002 (1, 3)
	WMV	0.53 ± 0.06	0.52 ± 0.05	0.49 ± 0.06	F = 4.878/*p* = 0.010 (1, 3)
Precuneus	MWF	6.99 ± 1.23	6.49 ± 1.32	5.68 ± 1.35	F = 7.712/*p* = 0.001 (1, 3)
	GMV	0.33 ± 0.03	0.31 ± 0.03	0.29 ± 0.04	F = 7.488/*p* = 0.001 (1, 3)
	WMV	0.26 ± 0.03	0.23 ± 0.03	0.32 ± 0.03	F = 13.292/*p* < 0.001 (1, 2), (1, 3)
Thalamus	MWF	7.11 ± 1.09	6.98 ± 1.21	6.07 ± 1.31	F = 6.617/*p* = 0.002 (1, 3),(2, 3)
	GMV	0.37 ± 0.04	0.36 ± 0.04	0.33 ± 0.05	F = 4.516/*p* = 0.014 (1, 3), (2, 3)
	WMV	0.27 ± 0.04	0.25 ± 0.03	0.24 ± 0.04	F = 4.017/*p* = 0.021 (1, 3)

The data are listed as the mean ± standard deviation. The units are % for MWF and cm^3^ for both the GMV and the WMV.The ANOVA test (F/*p* values *) was performed to compare the MRI measures between the three participant groups with the Scheffé post hoc test. The results of the post hoc tests are listed as the significant difference between the three subject groups (1, 2, 3), between the CN and AD groups (1, 3), between the CN and MCI groups (1, 2), and between the MCI and AD groups (2, 3).

**Table 3 diagnostics-12-00446-t003:** Results of correlation analysis between MRI measures in specific brain areas and age or MMSE scores.

ROI	Regressors	MWF		GMV		WMV	
Corpus callosum	Age	−0.565	<0.0001	−0.178	0.0905	−0.361	0.0004
	*adjMMSE	0.349	0.0007	0.281	0.0069	0.338	0.0011
Cingulate gyrus	Age	−0.537	<0.0001	−0.400	0.0001	−0.467	<0.0001
	*adjMMSE	0.235	0.0248	0.276	0.0081	0.365	0.0004
Cingulum	Age	−0.481	<0.0001	−0.474	<0.0001	−0.459	<0.0001
	*adjMMSE	0.151	0.1519	0.367	0.0004	0.353	0.0006
Hippocampus	Age	−0.429	<0.0001	−0.545	<0.0001	−0.468	<0.0001
	*adjMMSE	0.251	0.0165	0.539	<0.0001	0.421	<0.0001
Middle temporal gyrus	Age	−0.462	<0.0001	−0.429	<0.0001	−0.446	<0.0001
	*adjMMSE	0.186	0.0771	0.442	<0.0001	0.322	0.0019
Parahippocampal gyrus	Age	−0.371	0.0003	−0.546	<0.0001	−0.456	<0.0001
	*adjMMSE	0.248	0.0179	0.544	<0.0001	0.423	<0.0001
Pons	Age	−0.321	0.0018	−0.355	0.0005	−0.218	0.0366
	*adjMMSE	0.252	0.0161	0.367	0.0003	0.305	0.0032
Precuneus	Age	−0.481	<0.0001	−0.517	<0.0001	−0.496	<0.0001
	*adjMMSE	0.263	0.0117	0.424	<0.0001	0.406	0.0001
Thalamus	Age	−0.487	<0.0001	−0.417	<0.0001	−0.204	0.0509
	*adjMMSE	0.234	0.0257	0.242	0.0209	0.262	0.0122

The data are listed as Pearson’s correlation coefficient with *p*-values, except for *adjMMSE which are results of partial correlation analysis between MMSE scores and MRI measures with adjustment for age. MWF, apparent myelin water fraction; GMV, gray matter volume; WMV, white matter volume; ROI, region-of-interest; MMSE, Mini-Mental State Examination.

**Table 4 diagnostics-12-00446-t004:** Results of the added value analyses of MRI measures using a receiver operating characteristic (ROC) curve analysis for group classification.

MRI Measures	CN vs. MCI				CN vs. AD				MCI vs. AD			
	SE	SP	AUC	*p*	SE	SP	AUC	*p*	SE	SP	AUC	*p*
Hippo GMV	50.00	82.76	0.691	0.0058	86.21	80.65	0.883	<0.0001	61.29	84.37	0.771	<0.0001
Hippo WMV	71.87	65.52	0.659	0.0260	93.10	70.97	0.860	<0.0001	77.42	68.75	0.740	0.0001
CC MWF	75.00	41.38	0.517	0.8222	86.21	67.74	0.799	<0.0001	70.97	84.37	0.779	<0.0001
GMV + MWF	50.00	82.76	0.691	0.0056	93.10	77.42	** *0.898* **	<0.0001	70.97	87.50	** *0.803* **	<0.0001
GMV + WMV	84.37	55.17	0.690	0.0063	89.66	77.42	0.888	<0.0001	67.74	81.25	0.779	<0.0001
WMV + MWF	71.87	62.07	0.650	0.0367	96.55	70.97	** *0.891* **	<0.0001	70.97	81.25	** *0.794* **	<0.0001
GMV + WMV + MWF	84.37	55.17	0.690	0.0063	96.55	77.42	** *0.905* **	<0.0001	70.97	84.37	** *0.812* **	<0.0001

Data are listed as results of the receiver operating characteristic (ROC) analysis with Sensitivity (SE), Specificity (SP), area under the ROC curve (AUC), and the *p*-value. MRI measure values were obtained from hippocampus (Hippo) for gray matter volume (GMV), hippocampus (Hippo) for white matter volume (WMV), and corpus callosum (CC) for myelin water fraction (MWF) based on the ROC analysis of each MRI measure at each brain area (Appendix A).

## Data Availability

Data will be provided upon request.

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
