# Peer review of "Myelin-Weighted Imaging Presents Reduced Apparent Myelin Water in Patients with Alzheimer’s Disease"

_diagnostics, 2022, doi:10.3390/diagnostics12020446_

Round 1

Reviewer 1 Report

This prospective study evaluated the myelin changes in patients with Alzheimer’s disease (AD) and mild cognitive impairment (MCI) by using MR myelin water fraction (MWF) and investigated the associations of MWF with MMSE score and age, through the voxel-based and ROI-based analyses. In addition, authors performed the ROC analyses to compare the diagnostic values of different combinations of gray matter volume (GMV), white matter volume (WMV) and MWF. The results are well described and the method seems valid, which is easy to follow. However, the manuscript is not organized very well. And, there are several concerns regarding to the study I would like to put forward as below:

  1. As the P value of sex among three groups was nearly to 0.05, it would be better to induce the sex as covariate nuisance in all your voxel-based statistical models, not just the models of GMV and WMV.
  2. In the ROI-based analysis, as you selected several ROIs for comparisons, you need to perform the multiple comparison correction for this (e.g. FDR or Bonferroni method)
  3. In the voxel-based MWF regression models against MMSE, I would like to recommend you to reconstruct the general linear models separately in CN, MCI and AD, not just roughly using the whole cohort.
  4. For the comparisons among different AUC in different parameter combinations, you also need to generate the hypothesis test and use the Z test for comparing the areas of the ROC.
  5. Finally, in the “Discussion” section, it would be better to explain your result in the same order as you describe them in “Results” section.

Reviewer 2 Report

Lim and colleagues used a novel 3D ViSTa-GraSE sequence to map apparent myelin water fraction in Alzheimer and mild cognitive impairment elderly patients while comparing them to cognitively normal group. The results are interesting.

Minor comments that need to be addressed are listed below:

Abstract:  Spell out AUC=area under the curve and if you add the ROC (receiver operating characteristic) please explain this abbreviation too.

Intro: 1st sentence you need to be more specific in your description of the neuron and the oligodendrocyte/schwann cell. Same with demyelination (3rd sentence). Other than that, introduction is on point.

M&M: line 87 – Secondly,

Line 98 – At first or firstly,

Line 132 – Maybe a brief explanation for the scaling factor of 0.763561 would be fitting.

Discussion: Line 368 – The results of the current study confirm the findings of older ones, not the other way.

Line 387 – Progression

Line 396 – Consequently or as a consequence,

Line 409 – a promising

Author Response

Response to Reviewer 2 Comments

Lim and colleagues used a novel 3D ViSTa-GraSE sequence to map apparent myelin water fraction in Alzheimer and mild cognitive impairment elderly patients while comparing them to cognitively normal group. The results are interesting.

Minor comments that need to be addressed are listed below:

Point 1.

Intro: 1st sentence you need to be more specific in your description of the neuron and the oligodendrocyte/schwann cell. Same with demyelination (3rd sentence). Other than that, introduction is on point.

Response : We added these sentenses in the introduction

  • In a nervous system, a neuron is a cell that sends signals to other cells through fragile and thin axons while a glial cell forms a membranous sheath called myelin, which surrounds and eventually insulates, axons [1]. The oligodendrocytes and schwann cells are specialized glial cells that produce the myelin sheaths of the central nervous system and peripheral nervous system, respectively. The Oligodendrocytes myelinate multiple axons in central nervous system, while schwann cells myelinate a single axon in peripheral nervous system.
  • Demyelination is the condition in which preexisting myelin sheaths are damaged and loss without axonal damage. It can result from various medical conditions such as viral infection, inflammatory process, metabolic dysfunctions.

Point 2.

Line 132 – Maybe a brief explanation for the scaling factor of 0.763561 would be fitting.

Response : We added this sentence about the scaling factor

The scaling factor was 0.763561 which was considered to the difference of TR and TE for the ViSTa-GraSE image and the corresponding reference image and the exponential signal decay effects with T1 and T2* relaxation times.

Other points

Abstract: Spell out AUC=area under the curve and if you add the ROC (receiver operating characteristic) please explain this abbreviation too.

M&M: line 87 – Secondly,

Line 98 – At first or firstly,

Discussion: Line 368 – The results of the current study confirm the findings of older ones, not the other way.

Line 387 – Progression

Line 396 – Consequently or as a consequence,

Line 409 – a promising

Response : We've revised all the words what you suggested. Thank you.

Round 2

Reviewer 1 Report

Author's modification is basically as required.